# Association between Exposure to Domestic Violence during Childhood and Depressive Symptoms in Middle and Older Age: A Longitudinal Analysis in China

**DOI:** 10.3390/bs13040311

**Published:** 2023-04-05

**Authors:** Hui Lv, Haomiao Li

**Affiliations:** 1Institutes of Health Central Plains, Xinxiang Medical University, Xinxiang 453003, China; 2School of Political Science and Public Administration, Wuhan University, Wuhan 430072, China

**Keywords:** domestic violence, depressive symptoms, parental conflict, corporal punishment

## Abstract

Exposure to domestic violence (EDV) is a constant threat to social stability and global solidarity and may be associated with an increased risk of depression in later life. This study assessed the association between EDV during childhood and depressive symptoms in middle and older age. A total of 10,521 respondents obtained from the China Health and Retirement Longitudinal Study were enrolled in our analysis. Depressive symptoms were measured using the 10-item form of the Center for Epidemiological Studies Depression (CES-D) scale, and EDV included parental conflict and corporal punishment. A random-effects linear regression was used to assess associations. The results showed positive relationships between “not very often” (β = 0.862; 95% CI:0.512 to 1.211; *p* < 0.001), “sometimes” (β = 1.692; 95% CI:1.227 to 2.158; *p* < 0.001) and “often” (β = 2.143; 95% CI:1.299 to 2.987; *p* < 0.001) in parental conflict and the CES-D scores, compared with that of those reported “never” in parental conflict. Similarly, positive relationships between “sometimes” (β = 0.389; 95% CI:0.091 to 0.687; *p* = 0.011) and “often” (β = 1.892; 95% CI:1.372 to 2.413; *p* < 0.001) in corporal punishment and the CES-D scores were observed. EDV is associated with an increased risk of depression in later life. Future research could develop interventions that target EDV and explore the mechanisms in China to further decrease lifetime depression risk and improve the population’s mental health.

## 1. Introduction

Exposure to domestic violence (EDV) is defined as “being directly victimized by their caregivers and/or witnessing violence between their caregivers in the home” [1], which has been a global public health concern [2]. In a previous study based on a large sample of 18,341 adolescents in China, the lifetime prevalence of witnessing family violence ranged from 8.3% to 41.4% [3]. A meta-analysis showed that the lifetime prevalence of child abuse in mainland China was approximately 36.6%, which is significantly higher than the international estimate of 17.7% [4].

Domestic violence poses a constant threat to social stability and global solidarity [5]. In addition, EDV has been proved to be linked to lifetime psychosocial difficulties and physical and mental health problems [6,7]. On the one hand, EDV has negative effects on adolescents’ psychological health. Some previous studies indicated that internal and external behavioral problems and post-traumatic stress disorder (PTSD) are more prevalent among children and adolescents who have witnessed domestic violence and children suffering from abuse [8,9]. On the other hand, the negative effects of EDV exist not only in childhood and adolescence but also in adulthood and even middle and older age. For example, a study found that adolescents who experienced childhood abuse had more psychological problems such as depression and suicide attempts in adulthood, and had higher odds of health and behavioral problems such as eating disorders, drug and alcohol abuse, sexual infections, risky sexual behavior, and crime [10]. People who have experienced abuse and witnessed marital violence tend to have less secure attachments to their partners, less conflict resolution, and are more likely to be victims or perpetrators of partner violence, which further affects the quality of their intimate relationships and their physical and mental health [11]. These negative effects often persist until middle age and even in older age [12,13].

As previous studies have indicated, one of the most significant risks associated with EDV is depressive symptoms. To date, the associations between EDV and depressive symptoms among children and adolescents have been widely studied, especially in Western studies. Nevertheless, most studies have concentrated on children or adolescents’, rather than older adults’, depression levels. Life course perspectives suggest that biological and social events that occur during any life period can have subsequent and lifelong effects on population health [14]. The lifetime consequences of EDV, especially in middle-aged and older populations, have not been well studied. In particular, despite its high prevalence rate, the lifetime impact of EDV on depressive symptoms has not been well studied in China due to its acceptance rooted in Chinese culture [15].

Therefore, based on a Chinese sample, the present study aimed to examine the associations between EDV in childhood and depressive symptoms among older adults using a longitudinal design. The hypothesis of this study is that EDV in childhood is positively associated with an increased risk of depressive symptoms in older age (45 years and above).

## 2. Materials and Methods

### 2.1. Participants

The data for this study were obtained from the China Health and Retirement Longitudinal Study (CHARLS), which was conducted by the National School of Development of Peking University. CHARLS collected microdata from middle-aged and older Chinese individuals, aged ≥ 45 years, in 2011, 2013, 2015, and 2018, with multistage stratified probability-proportional-to-size sampling [16]. In addition, CHARLS conducted a life history survey in 2014, collecting information on residence, migration, family status, education, health and healthcare, wealth, and work history during childhood and adulthood from all live respondents in the first two surveys.

A total of 20,544 individuals were recruited in the CHARLS wave of 2014. We matched these data with those of the CHARLS baseline survey (2011) and the three follow-up assessments (2013, 2015, and 2018). We excluded some samples from this study based on the following criteria: (1) respondents younger than 45 years at the baseline survey and (2) respondents who did not respond whether they suffered from EDV during childhood. A total of 10,521 respondents were enrolled in the analysis of the associations between EDV and depressive symptoms.

### 2.2. Variables

#### 2.2.1. Dependent Variable

The respondents’ depressive symptoms were measured using the 10-item form of the Center for Epidemiological Studies Depression (CES-D) scale, with a response scale ranging from 0 (rarely or never) to 3 (most or all the time). The summed scores ranged from 0 to 30, with higher scores indicating negative feelings and a higher depressive symptom severity [17].

#### 2.2.2. Independent Variables

The independent variable in this study was two types of EDV, including parental conflict and corporal punishment. Parental conflict was measured using the question, “Have your father/mother ever beat up your mother/father?” Corporal punishment was measured using the question, “When you were growing up, did your female/male guardian ever hit you?” Both questions were responded to with a scale of 0 to 3 (0 = Never; 1 = Occasionally; 2 = Sometimes; 3 = Often).

#### 2.2.3. Covariates

The covariates in this study were identified according to previous studies, including age, gender, educational level, marital status, household registration (hukou) status, rural or urban residence, current smoking status, current drinking status, current work status, public health coverage, number of chronic diseases, and self-rated health.

### 2.3. Statistical Analysis

The mean ± standard deviation and the frequency (percentages) were used to describe the respondents’ characteristics. In this longitudinal analysis, smooth curve fitting for mean CES-D scores across different survey waves categorized by parental conflict and corporal punishment were generated based on general additive models (GAMs) while adjusting for all covariates, in order to visualize and compare the level and trajectory of CES-D scores between respondents with different EDV.

Random-effects linear regression was performed to assess the association between EDV and depressive symptoms with coefficients (β) and 95% confidence intervals (CI), and positive values of coefficients indicated positive associations. The calculation formula is as follows:$$\mathrm{LnDEP}=\beta_{0}+\beta_{1}X_{i}+\sum \beta_{j}COV_{i}+\varepsilon_{i}.$$

DEP indicates severity of depressive symptoms; X indicates level of EDV (including parental conflict and corporal punishment); i indicates individual; COV indicates covariates; $\beta_{0}$ is a constant term; $\beta_{1}$ is the parameter to be estimated; and $\varepsilon_{i}$ is a random error term.

To check the robustness of the results, we conduct two additional analyses. First, to avoid statistical test performance reduction and bias owing to the direct exclusion of missing values, multiple imputations based on five replications and a chained equation approach were performed to repeat the main analysis. Second, the dependent variables of this study (EDV) remained constant in the four survey waves, hence fixed-effects models were not suitable. Therefore, we performed generalized estimating equations (GEE) to repeat the analysis, which could then be applied to analyze data with normal, binomial, and Poisson distribution. Moreover, longitudinal data in repeated-measures design that do not meet the ANOVA conditions can be analyzed using GEE [18].

We further conducted two subgroup analyses for gender (female/male) and place of residence (urban/rural). Gender differences in trauma experience and impact have been reported in many previous studies but mainly in high-income countries [19]. Moreover, differences in the speed of urban and rural development, education levels of the populations, and legal systems in the areas may also affect the presence and severity of depressive symptoms. Many studies have indicated that depression risks as well as impacting factors vary in gender and residence in China [20,21].

The *p*-values were two-tailed and statistical significance was set at an alpha level of 0.05. Data were analyzed using Stata (version 15) and R version 3.6.3 (R Foundation for Statistical Computing, Vienna, Austria).

### 2.4. Ethics Approval and Consent to Participate

The Biomedical Ethics Review Committee of Peking University approved CHARLS, and all participants were required to provide written informed consent. The ethical approval number was IRB00001052-11015.

## 3. Results

The demographic characteristics of the sample (N = 10,521) in 2011 are presented in Table 1. The mean age of the respondents was 58.86 years old, 48.22% were male, and 51.78% were female. Most of the respondents were married (86.73%), with less than lower secondary education level (86.59%), covered by public health insurance (91.34%), with agricultural hukou (77.02%), and working (67.36%). More than half of the respondents (61.87%) lived in rural areas. Less than one-third of the respondents were drinking (32.16%) and smoking (29.31%) (in the survey year). For health status, 23.68%, 47.78%, and 28.54% of the respondents rated themselves as bad, average, and good, respectively. There were 1322 (12.57%), 688 (6.54%), and 184 (1.75%) respondents who occasionally, sometimes, and often witnessed parental conflict during childhood, respectively; and 3412 (32.43%) 2546 (24.20%), and 597 (5.67%) participants who occasionally, sometimes, and often suffered from corporal punishment, respectively.

We compared mean CES-D scores among groups with different EDV (Table 2), and the highest mean CES-D scores were observed in all survey waves among respondents who “Often” witnessed parental conflict and/or experienced corporal punishment. Smooth curve fitting for the trajectory of CES-D scores across groups with different EDV (Figure 1) also revealed a higher risk of depressive symptoms among respondents who “Often” witnessed parental conflict and/or experienced corporal punishment, with all the covariates adjusted.

We explored the effects of parental conflict and corporal punishment exposure in childhood on CES-D scores in middle and older age based on random-effects linear regression models (Table 3). With all the covariates adjusted, the results of random-effects linear regressions showed positive relationships between “Occasionally” (β = 0.862; 95% CI:0.512 to 1.211; *p* < 0.001), “Sometimes” (β = 1.692; 95% CI:1.227 to 2.158; *p* < 0.001), and “Often” (β = 2.143; 95% CI:1.299 to 2.987; *p* < 0.001) witnessing parental conflict and the CES-D scores, compared with those of who responded “Never” witnessing parental conflict. Similarly, positive relationships between “Sometimes” (β = 0.389; 95% CI:0.091 to 0.687; *p* = 0.011) and “Often” (β = 1.892; 95% CI:1.372 to 2.413; *p* < 0.001) experiencing corporal punishment and the CES-D scores were observed, compared with those who responded as “Never” experiencing corporal punishment.

The results of the robustness check were presented in Table 4. The results of parental conflict were highly consistent with the main analysis (Table 3), both using data after imputation and with a GEE model. For corporal punishment, the findings differed only slightly from those of the main analysis. Using imputed data, the association between “Sometimes” in corporal punishment and the CES-D scores was still positive but not statistically significant. This may be associated with the statistical test performance reduction owing to the direct exclusion of missing values. Through GEE, all frequencies of corporal punishment were positively associated with increased risk of depressive symptoms compared with that of those responded “Never”, whereas this positive association was not significant in the main analysis (Table 3). This may be associated with the difference of parameter estimation between a random-effects model and GEE, and also may be associated with the existence of omitted variables. In term of this, we further tested endogeneity through an instrumental variables (IV) model. The family’s socioeconomic status was introduced as the IV, and two stage least square (2SLS) models were applied. The results also showed that both parental conflict (β = 15.59, *p* < 0.001) and corporal punishment (β = 10.61, *p* < 0.001) were positively associated with CES-D scores. Though slight inconsistent statistical significances existed, the directions of all the associations were consistent. The above robustness check results reinforced the main conclusions of this study.

In the subgroup analysis (Table 5), significantly positive associations between “Occasionally”, “Sometimes”, and “Often” in parental conflict and an increased risk of depressive symptoms were observed in all the subgroups, including male, female, rural and urban respondents. For corporal punishment, the effects of “Occasionally” exposure were not significant, while that of “Often” exposure was significant in all subgroups. The significance of the association between “Sometimes” in corporal punishment and depressive symptoms varied in different subgroups, and were significant among male and rural respondents, but not among female and urban respondents.

## 4. Discussion

From a life course perspective, this study aimed to explore the associations between EDV during childhood and depressive symptoms in middle and older age (≥45 years old). We found that EDV during childhood, including parental conflict and corporal punishment, was closely related to an increased risk of depressive symptoms. We further explored these associations in different genders and residences, and found that witnessing parental conflict and frequent exposure to corporal punishment were consistently associated with an increased risk of depressive symptoms in all subgroups. As indicated by the life course perspective [22] and resiliency theory [23], childhood experiences have long-term effects on mental health in later life. Early childhood is a key period of rapid brain growth and cognitive development [24], during which social, family, and environmental factors can affect children’s health throughout life. Some previous studies indicated that witnessing and suffering from domestic violence have been shown to be associated with a variety of social and psychological problems, including detached relationships with parents, physical health sequelae, high-risk sexual behaviors, impairment of emotional regulation [13,25,26], and even suicide [27], yet these studies concentrated more on children and adults, rather than older populations. Moreover, these associations have not been fully explored in China.

Our study further proved the lifetime effects of EDV in China, focusing on depressive symptoms. In the Chinese cultural context, children’s feelings are always neglected when domestic violence occurs between parents because of the widespread belief that children should not be concerned about adult affairs [28]. In addition, many Chinese parents still believe that “spare the rod, spoil the child” and “beating is caring, and scolding is loving” [4]. Hence, Chinese society is more likely to accept the abusive behaviors of parents who are trying to teach their children at home [29]. This could also explain why occasional corporal punishment was not significantly associated with CES-D scores in some subgroups. Nevertheless, corporal punishment is consistently harmful to mental health. Childhood EDV makes people more vulnerable to depression when confronted with life stressors [30]. Especially in older age, when reflecting on their lives, older adults may recall old traumas which function as new stressors. In concordance with the aging processes, such as deteriorating health and cognitive function, older adults may not be able to cope with these old traumas and may become depressed [30].

Other potential mechanisms linking EDV and depression have also been explored in previous studies. For instance, Yrondi et al. demonstrated that self-esteem mediates the effect of childhood physical abuse on the intensity of depressive symptoms in a geriatric population [31]. In a cross-sectional study by Wielaard et al., a smaller social network and feelings of loneliness mediate the association between childhood abuse and early onset depression among older adults [32]. Jardim et al. indicated that childhood maltreatment is associated with geriatric depression and is mediated by personality factors, including neuroticism and extraversion as complete mediators, agreeableness and conscientiousness as partial mediators, while openness is not a mediator [33]. Clinical mechanisms, such as involvement of the hypothalamus-pituitary-adrenal (HPA) axis and, consequently, cortisol metabolism and stress reactivity [34], higher levels of inflammatory markers [35], increased salivary cortisol, and plasma ACTH concentrations [36], have also been reported in some previous studies. However, the mechanisms between early EDV and lifetime risk of depression require further exploration.

Our findings indicated that the domestic violence prevention measures should be implemented immediately to help victims fend off the adverse impacts of domestic violence, which are persistent and even lifelong. In China, the government should perfect the relative laws to prevent domestic violence. More importantly, this social issue should be solved with the efforts of the whole society. Education should be enhanced for both parents and children, to construct a harmonized family atmosphere. Social media should conduct vigorous propaganda about the illegitimacy of domestic violence and its negative effects. Parents’ understanding of the appropriate educational concepts and educational methods should be cultivated.

One strength of this study is the large sample size used to explore the association between EDV including different frequencies of parental conflict and corporal punishment, and depressive symptoms, which has rarely been measured in previous studies in China. Additionally, we performed several sensitivity and subgroup analyses, which yielded robust and consistent results. Nevertheless, this study had some limitations. First, the data were collected retrospectively and were therefore subject to recall bias. Second, although a longitudinal analysis was conducted, the causal effects of EDV on depression could not be justified in our study, and the mechanism could not be analyzed. Third, only participants who responded to all survey waves were included in this study, and some participants were excluded from the data analysis due to missing data. Therefore, the representativeness of the study’s findings should be interpreted with caution. Further life-course studies are needed to evaluate the causal effects of EDV on depression and the mechanisms should be explored more in depth.

## 5. Conclusions

The population who witnessed or suffered from domestic violence, including parental conflict and corporal punishment, is more vulnerable to depressive symptoms among middle-aged and older individuals. Domestic violence endangers public health and social stability. We call for the Chinese government to develop effective interventions to address domestic violence epidemics in China in a timely manner. It is also necessary to strengthen coordination and set up a social support and assistance network consisting of public, prosecutors, law, departments, mass groups and self-government groups. The roles of government, schools, and social media should be enhanced. To further shed light on this issue, future research could develop interventions that target EDV and explore the mechanisms in China to further improve the population’s mental health, social stability, and global solidarity.

## Figures and Tables

**Figure 1 behavsci-13-00311-f001:**
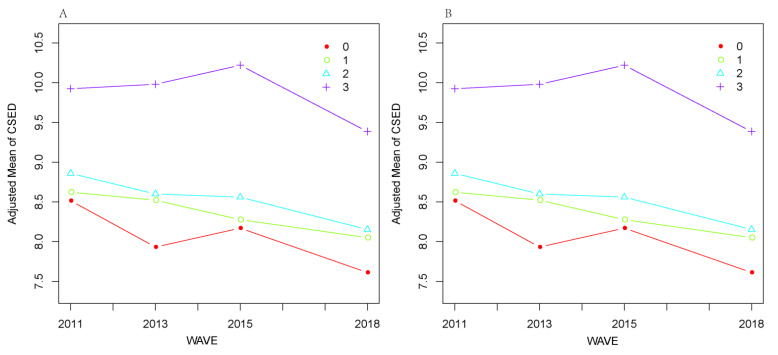
Smooth curve fitting for mean CES-D scores across different survey waves categorized by parental conflict and corporal punishment: (**A**) parental conflict; (**B**) corporal punishment. 0, Never; 1, Occasionally; 2, Sometimes; 3, Often.

**Table 1 behavsci-13-00311-t001:** Demographic characteristics and EDV status of the sample (2011).

	Mean ± SD/N (%)
Age	58.86 ± 10.50
Gender	
Male	5071 (48.22%)
Female	5445 (51.78%)
Marital status	
Divorced or widowed	958 (13.27%)
Married	6262 (86.73%)
Education levels ^a^	
Less than lower secondary	9088 (86.59%)
Upper secondary and vocational training	1107 (10.55%)
Tertiary	300 (2.86%)
Public health insurance coverage ^b^	
Not covered	617 (8.66%)
Covered	6505 (91.34%)
Hukou status ^c^	
Agricultural	5559 (77.02%)
Non-agricultural	1624 (22.50%)
Other	35 (0.48%)
Residence	
Rural	4467 (61.87%)
Urban	2753 (38.13%)
Current work status	
Not working	2330 (32.64%)
Working	4808 (67.36%)
Alcohol intake	
Do not drink	4909 (67.84%)
Drink	2327 (32.16%)
Smoking status	
Do not smoke	4882 (70.69%)
Smoke	2024 (29.31%)
Number of chronic diseases ^d^	1.00 ± 1.23
Self-rated health	
Bad	1698 (23.68%)
General	3426 (47.78%)
Good	2046 (28.54%)
Parental conflict	
Never	8327 (79.15%)
Occasionally	1322 (12.57%)
Sometimes	688 (6.54%)
Often	184 (1.75%)
Corporal punishment	
Never	3966 (37.70%)
Occasionally	3412 (32.43%)
Sometimes	2546 (24.20%)
Often	597 (5.67%)

Note: SD, standard deviation. ^a^ Education levels were classified by a simplified version of the 1997 International Standard Classification of Education codes. ^b^ Public health insurance includes Urban Employee Medical Insurance, Urban Resident Medical Insurance, New Cooperative Medical Insurance, Urban and Rural Resident Medical Insurance, Government Medical Insurance, Medical Aid or other government insurance plan. ^c^ Hukou status indicates the respondent’s hukou place and is a special identifier in China. Hukou status affects many aspects of life in China such as buying a house, buying a car, children’s school enrollment, and other welfare. ^d^ Chronic diseases included hypertension, diabetes, dyslipidemia, heart disease, stroke, cancer, chronic lung disease, digestive disease, liver disease, kidney disease, and arthritis.

**Table 2 behavsci-13-00311-t002:** Depressive symptoms across groups with different EDV.

	2011	2013	2015	2018
**Parental conflict**				
Never	5.58 ± 6.36	6.05 ± 6.01	7.21 ± 6.43	7.25 ± 6.66
Occasionally	5.96 ± 6.66	6.63 ± 6.43	7.96 ± 6.34	8.41 ± 6.87
Sometimes	6.62 ± 7.14	7.45 ± 6.52	8.86 ± 6.63	8.97 ± 7.09
Often	7.00 ± 6.57	8.36 ± 6.59	9.82 ± 7.18	9.40 ± 7.58
**Corporal punishment**				
Never	5.89 ± 6.47	6.28 ± 6.05	7.25 ± 6.36	7.33 ± 6.81
Occasionally	5.39 ± 6.33	6.01 ± 6.06	7.28 ± 6.35	7.52 ± 6.64
Sometimes	5.76 ± 6.48	6.33 ± 6.16	7.58 ± 6.54	7.62 ± 6.62
Often	6.29 ± 7.01	7.15 ± 6.75	9.29 ± 7.23	8.73 ± 7.52

Note: EDV, exposure to domestic violence. Mean ± SD are presented in the table.

**Table 3 behavsci-13-00311-t003:** Association between EDV and depressive symptoms based on random-effects linear regression.

	Parental Conflict		Corporal Punishment	
	β (95%CI)	*p* Value	β (95%CI)	*p* Value
EDV (Ref. Never)				
Occasionally	0.862 (0.512, 1.211)	<0.001	0.224 (−0.053, 0.501)	0.114
Sometimes	1.692 (1.227, 2.158)	<0.001	0.389 (0.091, 0.687)	0.011
Often	2.143 (1.299, 2.987)	<0.001	1.892 (1.372, 2.413)	<0.001
Age	−0.010 (−0.022, 0.001)	0.084	−0.011 (−0.023, 0.000)	0.058
Gender (Ref. Male)	0.068 (−0.214, 0.349)	0.636	0.081 (−0.201, 0.363)	0.574
Education levels (Ref. Less than lower secondary)				
Upper secondary and vocational training	0.095 (−0.314, 0.504)	0.649	0.124 (−0.286, 0.534)	0.552
Tertiary	−0.107 (−0.956, 0.742)	0.804	−0.114 (−0.964, 0.737)	0.793
Marital status (Ref. Divorced or widowed)	0.274 (−0.046, 0.594)	0.093	0.259 (−0.062, 0.579)	0.114
Hukou status (Ref. Agricultural)				
Non-agricultural	−0.044 (−0.331, 0.244)	0.766	−0.046 (−0.334, 0.243)	0.756
Other	0.108 (−0.631, 0.847)	0.774	0.156 (−0.584, 0.895)	0.68
Residence (Ref. Rural)	−1.735 (−1.972, −1.499)	<0.001	−1.776 (−2.013, −1.539)	<0.001
Public health insurance coverage (Ref. Not covered)	−0.094 (−0.422, 0.233)	0.573	−0.091 (−0.419, 0.237)	0.588
Current work status (Ref. Not working)	−0.078 (−0.257, 0.100)	0.389	−0.082 (−0.261, 0.096)	0.366
Alcohol intake (Ref. Do not drink)	0.026 (−0.197, 0.249)	0.818	0.018 (−0.206, 0.241)	0.877
Smoking status (Ref. Do not smoke)	0.142 (−0.134, 0.419)	0.313	0.144 (−0.133, 0.421)	0.308
Number of chronic diseases	−0.033 (−0.105, 0.039)	0.37	−0.028 (−0.100, 0.045)	0.452
Self-rated health (Ref. Bad)				
General	0.205 (−0.013, 0.422)	0.065	0.216 (−0.002, 0.434)	0.052
Good	0.221 (−0.048, 0.490)	0.108	0.231 (−0.038, 0.501)	0.092

Note: CI, confidence interval; EDV, exposure to domestic violence.

**Table 4 behavsci-13-00311-t004:** Robustness check.

	Multiple Imputation		GEE	
	β (95%CI)	*p* Value	β (95%CI)	*p* Value
Parental conflict (Ref. Never)				
Occasionally	0.657 (0.387, 0.927)	<0.001	0.819 (0.439, 1.199)	<0.001
Sometimes	1.414 (1.053, 1.775)	<0.001	1.679 (1.157, 2.200)	<0.001
Often	2.001 (1.322, 2.681)	<0.001	1.944 (1.077, 2.811)	<0.001
Corporal punishment (Ref. Never)				
Occasionally	−0.109 (−0.322, 0.105)	0.318	0.296 (0.005, 0.588)	0.046
Sometimes	0.145 (−0.087, 0.377)	0.221	0.484 (0.165, 0.804)	0.003
Often	1.209 (0.808, 1.610)	<0.001	1.793 (1.216, 2.369)	<0.001

Note: CI, confidence interval; GEE, generalized estimating equations.

**Table 5 behavsci-13-00311-t005:** Subgroup analysis.

	Parental Conflict		Corporal Punishment	
	β (95%CI)	*p* Value	β (95%CI)	*p* Value
Male respondents (Ref. Never) ^a^				
Occasionally	0.670 (0.166, 1.174)	0.009	0.208 (−0.199, 0.614)	0.316
Sometimes	1.706 (1.014, 2.398)	<0.001	0.470 (0.033, 0.907)	0.035
Often	2.158 (0.909, 3.408)	0.001	1.826 (1.090, 2.561)	<0.001
Female respondents (Ref. Never) ^a^				
Occasionally	1.028 (0.544, 1.513)	<0.001	0.240 (−0.139, 0.62)	0.214
Sometimes	1.680 (1.049, 2.310)	<0.001	0.315 (−0.093, 0.723)	0.130
Often	2.058 (0.911, 3.206)	<0.001	1.960 (1.221, 2.699)	<0.001
Rural respondents (Ref. Never) ^b^				
Occasionally	0.757 (0.300, 1.215)	0.001	0.088 (−0.278, 0.455)	0.636
Sometimes	1.777 (1.165, 2.388)	<0.001	0.438 (0.045, 0.831)	0.029
Often	1.872 (0.808, 2.936)	0.001	1.889 (1.189, 2.589)	<0.001
Urban respondents (Ref. Never) ^b^				
Occasionally	0.997 (0.456, 1.538)	<0.001	0.411 (−0.012, 0.834)	0.057
Sometimes	1.576 (0.856, 2.295)	<0.001	0.313 (−0.144, 0.770)	0.180
Often	2.609 (1.208, 4.011)	<0.001	1.917 (1.140, 2.694)	<0.001

Note: CI, confidence interval. ^a^ All the covariates were adjusted except for gender. ^b^ All the covariates were adjusted except for residence.

## Data Availability

All the original data can be obtained from the official website of CHARLS (http://charls.pku.edu.cn/ (accessed on 5 January 2023)).

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
