# Peer review of "Association between Exposure to Domestic Violence during Childhood and Depressive Symptoms in Middle and Older Age: A Longitudinal Analysis in China"

_behavsci, 2023, doi:10.3390/bs13040311_

Round 1

Reviewer 1 Report

This is a very strong and well-written article that makes an important contribution to the literature. We clearly see that for middle-aged and older Chinese adults, a history of EDV puts individuals at risk of depression in their later years. Many people want to believe that after a traumatic abusive childhood, they can free themselves of the consequences. When I taught domestic violence and substance abuse college courses, I reminded students that we can take our troubled past right along with us and bear the consequences lifelong. Of course, this work with a Chinese sample is of special importance because it sheds light on cultural factors that impact our perceptions of our childhoods.

The Methods section is strong with a very large sample. The use of the CES-D scale is an appropriate tool. The statistical analyses are not my area of expertise and I hope another reviewer can be helpful here. Table 1 makes clear that there is a low rate of parental conflict with higher rates of corporal punishment. The Results section is complicated and geared toward readers with statistical expertise.

The next to the last paragraph in the article is very strong. It does a good job of reviewing the strengths and limitations of the work.

Really, the only issue I have with the article is that the authors have not yet offered specific recommendations in the Conclusion section. The authors write: "We call for the Chinese government to develop effective interventions to address domestic violence epidemics in China in a timely manner" (line #s 291-292). The authors should say much more about how to go about developing effective interventions. Who or what in the Chinese government can do this? Is there a ministry of health that can be responsible for this effort? Is some new type of organization needed to tackle this problem? What kinds of interventions should be explored? Is education about family violence something that needs to be included? Overall, the authors are well-qualified to recommend next steps and should not miss this opportunity.

Author Response

Response: We are grateful for the reviewer's constructive insights to our manuscript. Your suggestions are much valuable for improving the practical significance of this study. We have added more discussions about developing effective interventions. The details are as follows:

In the discussion section, we added a paragraph:

Our findings indicated that, the domestic violence prevention measures should be implemented immediately, to help victims fend off the adverse impacts of domestic violence, which is persistent and even life-long. In China, the government should perfect the relative laws to prevent domestic violence. More importantly, this social issue should be solved with the efforts of the whole society. Education should be enhanced for both parents and children, to construct harmonised family atmosphere. Social media should conduct vigorous propaganda about the illegitimacy of domestic violence and its negative effects. Parents should be cultivated the appropriate educational concepts and educational methods.

In the Conclusions section, we added:

We call for the Chinese government to develop effective interventions to address domestic violence epidemics in China in a timely manner. It is also necessary to strengthen coordination, and set up a social support and assistance network, consisting of public, prosecutors, law, departments, mass groups and self-government groups. The roles of government, schools and social media should be enhanced.  

Reviewer 2 Report

1. In the GEE model of Table 4 Robustness check , the regression coefficient for occasional encounters with corporal punishment is positively significant, which is inconsistent with the significance of the occasional encounter term in Table 3 Association between EDV and depressive symptoms based on random-effects linear regression . Is the article model sufficiently robust? An exhaustive explanation of the part of the article's robustness test where the significance is inconsistent with the underlying regression is recommended.

2. In Table 5 Subgroup analysis, the description " For corporal punishment, the effects of "Not very often" exposure were not significant" is inconsistent with the results of the urban in the subgroup analysis in Table 5. It is recommended that the subgroup analysis be redescribed in conjunction with Table 5.

3. In the first paragraph of 4. Discussion, the description "To the best of our knowledge, this is the first study to assess the associations between EDV during childhood and depressive symptoms in middle and older age in China." is too absolute. In fact, similar studies already exist in China.3.E.g. Li Yue, Lu JH. A study on the effect of childhood adversity on depression in older adults[J]. Journal of Population,2020,42(04):56-69.DOI:10.16405/j.cnki.1004-129X.2020.04.005.It is suggested that this sentence should be reworked.

4. Does the article consider the existence of omitted variables when only random-effects linear regression, robustness tests and subgroup analysis are performed? It is recommended that the article be endogeneity tested to further improve the structure of the article.

Author Response

  1. In the GEE model of Table 4 Robustness check, the regression coefficient for occasional encounters with corporal punishment is positively significant, which is inconsistent with the significance of the occasional encounter term in Table 3 Association between EDV and depressive symptoms based on random-effects linear regression. Is the article model sufficiently robust? An exhaustive explanation of the part of the article's robustness test where the significance is inconsistent with the underlying regression is recommended.

Response: We are grateful for the reviewer's constructive insights to our manuscript. The significance is inconsistent may be caused by the strengthens and limitations of different regression models. We added some explanations for these inconsistences. The details are as follows:

For corporal punishment, the findings differed only slightly from those of the main analysis. Using imputed data, the association between “Sometimes” in corporal punishment and the CES-D scores was still positive but not statistically significant. This may be associated with the statistical test performance reduction owing to the direct exclusion of missing values. Through GEE, all frequencies of corporal punishment were positively associated with increased risk of depressive symptoms compared with that of those responded “never”, whereas this positive association was not significant in the main analysis (Table 3). This may be associated with the difference of parameter estimation between random-effects model and GEE, and also may be associated with the existence of omitted variables. In term of this, we further tested endogeneity through instrumental variables (IV) model. The family’s socioeconomic status was introduced as the IV, and two stage least square (2SLS) models were applied. The results also showed that both parental conflict (β=15.59, P<0.001) and corporal punishment (β=10.61, P<0.001) were positively associated with CES-D scores. Though slight inconsistent statistical significances existed, the directions of all the associations were consistent. The above robustness check results reinforced the main conclusions of this study.

  1. In Table 5 Subgroup analysis, the description " For corporal punishment, the effects of "Not very often" exposure were not significant" is inconsistent with the results of the urban in the subgroup analysis in Table 5. It is recommended that the subgroup analysis be redescribed in conjunction with Table 5.

Response: We are grateful for the reviewer's constructive insights to our manuscript. In this study, the p-values were two-tailed and statistical significance was set at an alpha level of 0.05. Therefore, in Table 5, the effects of "Not very often" exposure were not significant in the urban subgroup (P=0.057), which is consistent for the description " For corporal punishment, the effects of "Not very often" exposure were not significant".

  1. In the first paragraph of 4. Discussion, the description "To the best of our knowledge, this is the first study to assess the associations between EDV during childhood and depressive symptoms in middle and older age in China." is too absolute. In fact, similar studies already exist in China.3.E.g. Li Yue, Lu JH. A study on the effect of childhood adversity on depression in older adults[J]. Journal of Population,2020,42(04):56-69.DOI:10.16405/j.cnki.1004-129X.2020.04.005.It is suggested that this sentence should be reworked.

Response: We apologize for this inappropriate statement. We deleted this sentence.

  1. Does the article consider the existence of omitted variables when only random-effects linear regression, robustness tests and subgroup analysis are performed? It is recommended that the article be endogeneity tested to further improve the structure of the article.

Response: We are grateful for the reviewer's suggestion. We added endogeneity test through instrumental variables model to handle this issue. The results were consistent. The details are as follows:

 This may be associated with the difference of parameter estimation between random-effects model and GEE, and also may be associated with the existence of omitted variables. In term of this, we further tested endogeneity through instrumental variables (IV) model. The family’s socioeconomic status was introduced as the IV, and two stage least square (2SLS) models were applied. The results also showed that both parental conflict (β=15.59, P<0.001) and corporal punishment (β=10.61, P<0.001) were positively associated with CES-D scores. Though slight inconsistent statistical significances existed, the directions of all the associations were consistent. The above robustness check results reinforced the main conclusions of this study.